# SAGE: Fast, Generalizable and Photorealistic 3D Human Reconstruction from a Single Image

## Abstract

In this paper, we present SAGE, a Large Human Reconstruction Model, that can produce a photorealistic 3D reconstruction of a human from a single image in less than 1 second. To support scalable model training, we first design an effective data generation pipeline to alleviate the shortage of available photorealistic 3D human data. In this pipeline, we follow two strategies. The first one is to leverage existing rigged assets and animate them with extensive poses from daily life. The second strategy is to utilize existing multi-camera captures of humans and employ fitting to generate more diverse views for training. These two strategies enable us to scale up to 100k assets, significantly enhancing both the quantity and the diversity of data for robust model training. In terms of the architecture, our framework is inspired by Large Reconstruction Models (LRMs) and extracts tokenized features from the input image and the estimated simplified human mesh (SMPL) without detailed geometry or appearance. A mapping network takes this tokenized information as conditioning and employs a cross-attention mechanism to iteratively enhance an initial feature representation. Ultimately, the output is a triplane representation that depicts the 3D human, while novel views are rendered using a standard ray marching method given a camera viewpoint. Extensive experiments on three benchmarks demonstrate the superiority of our approach, both quantitatively and qualitatively, as well as its robustness under diverse input image conditions.

## 1 Introduction

Photorealistic 3D human reconstruction has gained significant attention due to its wide applications, including virtual reality Mikhailova et al. (2024), telepresence Tu et al. (2024), and human-computer interaction Cork et al. (2024). All these applications require *fast* and *photorealistic* 3D human reconstruction, ideally from *a single image*. However, single-view 3D reconstruction is naturally an ill-posed problem Sinha & Adelson (1993), due to the ambiguity of the unseen viewpoints. To deal with this problem, existing single-view photorealistic 3D human reconstruction methods Ho et al. (2024); Zhang et al. (2024b) have explored incorporating human-related inductive bias. These methods first inject strong structural priors via parametric body models (*e.g.,* SMPL Loper et al. (2015) or SMPL-X Pavlakos et al. (2019)), and overlay advanced 3D representations (*e.g.,* NeRF Mildenhall et al. (2020) or 3D Gaussian Splatting Kerbl et al. (2023)) to achieve photorealistic rendering. Simultaneously, diffusion-based priors are employed to hallucinate the unseen side and back views, further alleviating view ambiguity and refining the final geometry and appearance Ho et al. (2024); Zhang et al. (2024b). Nevertheless, this class of methods requires slow optimization on a per-instance basis, limiting the scalability and broader applications.

In this paper, we propose **SAGE**, a **S**ingle-view 3D human reconstruction model that is f**A**st, **G**eneralizable, and photor**E**alistic. SAGE enables high-quality reconstruction in less than one second, as shown in Figure 1. The key of SAGE is the adoption of recently developed large-scale single-view reconstruction models, *i.e.,* LRMs Hong et al. (2024). However, building an LRM for 3D human reconstruction is non-trivial for two reasons. First, we have a limited amount of 3D human assets for training such models. Unlike general-category datasets, *e.g.,* Objaverse Deitke et al. (2023), which contains 800K shape instances, the current human datasets Yu et al. (2021); Shen et al. (2023); Han et al. (2023) usually only include a few thousand human instances. This scarcity of training data makes it very challenging to train an LRM for human reconstruction. Second, the current LRM architectures are designed for reconstruction of general-category objects, without

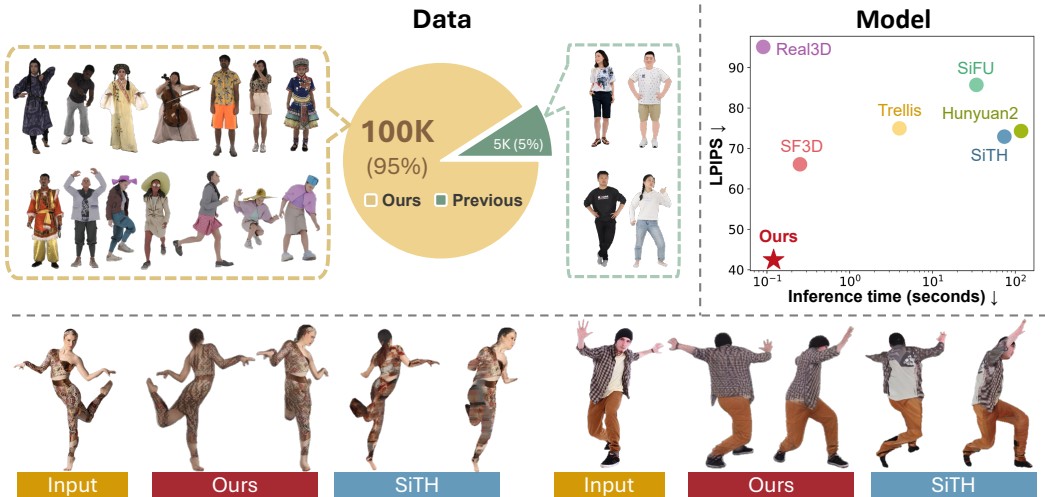

Figure 1: **Fast, generalizable and photorealistic 3D human reconstruction from a single image by the proposed approach, SAGE.** SAGE benefits from both our generated large-scale data and feed-forward model design. Our data generation pipeline expands training data by 20 times (top-left for visualization). With this data, SAGE achieves superior performance while maintaining *fast* inference among existing methods (top-right). Once trained, it is *generalizable* without the need for test-time fine-tuning or adaptation. Qualitative results show that SAGE produces more precise *photorealistic* reconstructions compared to the state-of-the-art SiTH method Ho et al. (2024) (bottom).

incorporating category-specific priors. However, in specific settings of high importance, like human reconstruction, we anticipate that a human-related prior will be beneficial.

To train an effective Large Human Reconstruction Model, we first scale up the training data. More specifically, we design a streamlined data generation pipeline, which introduces large-scale and high-quality human data. This data generation pipeline incorporates both synthetic and real data. The synthetic data benefits LRM training with its large scale, while the real data is more photorealistic and is well aligned with the data distribution in real applications. We generate 100K human instances in total, which is about 20 times larger than the combination of all existing datasets. This largely improves the reconstruction quality of humans with LRMs.

To build on this foundation, we adjust the LRM architecture for 3D human reconstruction by incorporating more human-related priors in the model. Concretely, SAGE leverages the parametric SMPL human mesh, robustly estimated by Goel et al. (2023), which provides a reasonably accurate yet coarse geometric prior to guide 3D human reconstruction.

Overall, our contributions are summarized as follows:

- We introduce SAGE, a generalizable large human reconstruction model that can reconstruct a 3D human from a single image in less than one second. We demonstrate that both data and model are crucial for extending the success of LRMs from general object reconstruction to the human domain.

- We design an effective data generation pipeline, which scales to 100k assets for our training dataset. Our pipeline considers both synthetic and real-world data, which significantly enhances both data quantity and diversity.

- We achieve significant improvements over the state of the art. Extensive experiments on multiple benchmarks demonstrate the superiority of SAGE both quantitatively and qualitatively, achieving over 40% relative LPIPS performance gain and over 90% relative F-Score gain when compared with state-of-the-art methods.

## 2 RELATED WORK

### 2.1 SINGLE-VIEW 3D HUMAN RECONSTRUCTION

Single-view 3D human reconstruction is a challenging task due to its inherent ill-posed nature. Early methods usually leveraged the mesh produced by parametric body models as the base body shape

and depicted detailed clothing via mesh offsets Alldieck et al. (2019b; 2018; 2019a) or adjustable garment templates Bhatnagar et al. (2019); Feng et al. (2023); Jiang et al. (2020). However, the constraint of the mesh topology limits the capability of these methods to model complex clothing (*e.g.,* dresses). Recent methods tend to utilize neural representations, *e.g.* NeRF, SDF, and 3DGS, given their flexibility with topology modeling Kwon et al. (2021); AlBahar et al. (2023); Corona et al. (2023); Xiu et al. (2022); Zheng et al. (2021); Chen et al. (2025); He et al. (2021); Liao et al. (2023); Xiu et al. (2023); Huang et al. (2024a); Chu et al. (2024); Yang et al. (2025); Wang et al. (2024); Lu et al. (2025); Zhuang et al. (2024). For improved texture quality, researchers refine human meshes generated through iterative optimization as a guiding signal, while simultaneously leveraging image priors from diffusion models Zhang et al. (2024a); Zhan et al. (2024); Pan et al. (2024); Li et al. (2025); Ho et al. (2024); Zhang et al. (2024b); Xue et al. (2024). For example, GTA Zhang et al. (2024a) combines encoding with a visual transformer and a decoder using cross-attention on triplane features. DIFu Song et al. (2023) introduces a depth-guided implicit function that uses front and hallucinated back depth maps projected into a 3D volume to provide voxel-level priors for better surface details. Pan et al. (2024) propose a framework built on 3DGS, which integrates a multi-view diffusion model and a latent reconstruction transformer, to recreate generalizable human views. SiTH Ho et al. (2024) decomposes this task into hallucination of images from unseen views with diffusion models and reconstruction by leveraging skinned body meshes. Weng et al. (2024) design a multi-stage framework to leverage the prior from large reconstruction and diffusion models. SIFU Zhang et al. (2024b) employs a side-view transformer coupled with the fitted SMPL-X mesh to map 2D features to 3D space, followed by the incorporation of priors via a diffusion model. The limitation of these approaches is the excessive reliance on diffusion priors. This results in significantly increased inference time and might lead to restricted generalization capabilities when the training data is limited. In this work, we tackle this task from a different perspective by developing a fast inference framework while significantly scaling up our training data.

## 2.2 SINGLE-VIEW 3D OBJECT RECONSTRUCTION

Early work for 3D object reconstruction from a single-view focused on category-specific approaches Kanazawa et al. (2018); Tatarchenko et al. (2019); Xu et al. (2019). Recently, researchers have explored building models that enable 3D reconstruction of general category objects from a single view. One group of approaches finetunes generative models to generate novel views and reduces the problem to multi-view reconstruction. A pioneering work in this space is Zero-1-to-3 Liu et al. (2023), which utilizes a viewpoint-conditioned diffusion model trained on synthetic datasets. Follow-up works focus on tackling the potential inconsistencies in the generated multi-view images Long et al. (2024); Shi et al. (2024); Liu et al. (2024). Even without inconsistencies, the final 3D representation needs to be derived by optimization given the generated views, which can be computationally expensive. Another line of work is building large-scale feed-forward models to directly learn 3D reconstruction from data Hong et al. (2024); Jiang et al. (2023); Boss et al. (2025); Zou et al. (2024); Wei et al. (2024); Szymanowicz et al. (2024). LRM Hong et al. (2024) is one of the pioneering works that uses a scalable transformer-based network and trains on large-scale 3D data to directly map the 2D image information to the 3D representation. Follow-up works improve the reconstruction quality by hallucinating multiple novel views Wang et al. (2025); Xu et al. (2024a); Tang et al. (2024); Xu et al. (2024b), designing better architectures Boss et al. (2025) or curating data of larger scale Jiang et al. (2025). To improve the generalization capabilities, Real3D Jiang et al. (2025) designs a self-training procedure to augment the training data with large-scale single-view images. SF3D Boss et al. (2025) further improves the network architecture, reducing memory consumption while enhancing the network's representational capacity for high-quality reconstruction. In this work, we design our framework based on LRMs and explore strategies to achieve high-quality and fast 3D reconstruction specifically in the human domain.

## 3 SAGE

In this section, we first introduce the model architecture of SAGE, and then present our data generation method for synthesizing large-scale human training data.

### 3.1 FRAMEWORK

As shown in Figure 2, following the LRM strategy Hong et al. (2024); Jiang et al. (2025); Boss et al. (2025), our framework achieves 3D reconstruction in a single feed-forward pass. Specifically, the

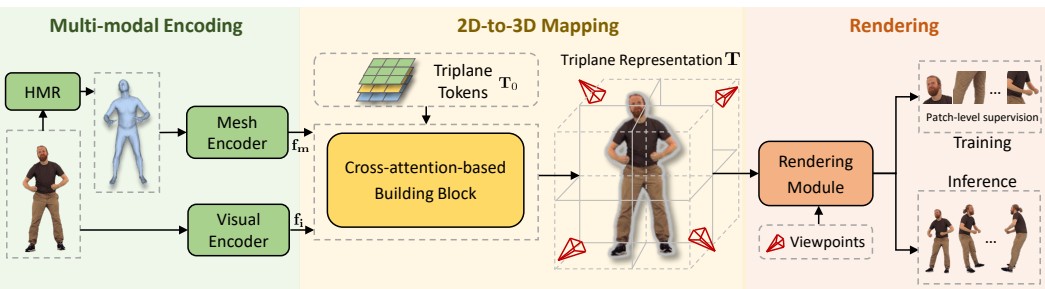

Figure 2: **SAGE network architecture**. Given a real-world input image, we first estimate its corresponding simplified human mesh. Image and mesh are fed into the multi-modal encoder to extract features which are utilized as the condition for the following mapping network. After that, a Transformer-based mapping network directly maps the features to the 3D triplane representation. From this triplane representation, our framework can render the 2D image given a camera viewpoint.

model consists of three main components, *i.e.,* the multi-modal encoding, the 2D-to-3D mapping, and rendering.

Concretely, given an image $I \in \mathbb{R}^{H \times W \times 3}$, we use a DINOv2 Oquab et al. (2024) encoder to tokenize it into feature tokens, denoted as $\mathbf{f_i} \in \mathbb{R}^{N_i \times d}$. Here $N_i$ is the number of visual tokens, and $d$ is the latent dimension. Specifically, we have $N_i = HW/p^2$, where $p$ is the patch size of DINOv2. Meanwhile, given the input image, we estimate the human surface Goel et al. (2023) in the form of the SMPL parametric model Loper et al. (2015), and we use PTv3 Wu et al. (2024) to tokenize the corresponding estimated human mesh. This additional input offers an initial estimate of the body pose and body surface for the person in the image. The output feature is denoted as $\mathbf{f_m} \in \mathbb{R}^{N_m \times d}$, while $N_m$ is the number of mesh tokens.

These encoded features are then fed into the mapping network to transform the image tokens $\mathbf{f_i}$ and mesh tokens $\mathbf{f_m}$ to the 3D triplane representation $\mathbf{T} \in \mathbb{R}^{3hw \times d}$, where $h$ and $w$ denote the spatial resolution of the triplane. This triplane representation $\mathbf{T}$ is a set of learnable tokens, serving as shared initialization for all inputs. Inspired by SF3D Boss et al. (2025), we implement our mapping network based on PointInfinity Huang et al. (2024b). This mapping network consists of multiple blocks, where each block updates the triplane tokens by fusing multi-modal condition information and performing triplane refinement, which is formulated as follows:

$$\mathbf{L}^l = \text{CrossAttn}(\text{Q} = \mathbf{f_i}||\mathbf{f_m}, \text{KV} = \mathbf{T}^l), \tag{1}$$

$$\mathbf{L}^l = \text{CrossAttn}(\text{Q} = \mathbf{L}^l, \text{KV} = \mathbf{f_i}||\mathbf{f_m}), \tag{2}$$

$$\mathbf{T}^{l+1} = \text{CrossAttn}(\text{Q} = \mathbf{T}^l, \text{KV} = \mathbf{L}^l), \tag{3}$$

where the initial token and the final output are denoted as $\mathbf{T}_0$ and $\mathbf{T}$, respectively. Then, we render images from the triplane following the standard ray marching method Hong et al. (2024); Jiang et al. (2025). We denote the process as $\hat{I}_\Phi = \pi(\mathbf{T}^*, \Phi)$, where $\pi$ denotes the rendering function, $\Phi$ is a target camera viewpoint, and $\hat{I}_\Phi$ is the image rendered under this target camera pose.

**Training objectives.** The whole framework is jointly trained under multiple objective functions, including RGB loss $\mathcal{L}_r$, mask loss $\mathcal{L}_m$ and LPIPS loss $\mathcal{L}_p$ as follows:

$$\mathcal{L} = \frac{1}{N} \sum_{n=1}^{N} \left( \mathcal{L}_r^n + \lambda_m \mathcal{L}_m^n + \lambda_p \mathcal{L}_p^n \right), \tag{4}$$

where $\lambda_m$ and $\lambda_p$ are loss weighting terms, and $N$ is the number of rendered views. The RGB $\mathcal{L}_r$ loss and LPIPS $\mathcal{L}_m$ loss focus on the quality of the rendered image from different perspectives. The mask loss $\mathcal{L}_m$ calculates the consistency between the accumulated density and the estimated mask. To reduce the GPU memory consumption, all loss terms are calculated on the patch level, which is extracted by a weighted sampling strategy based on the foreground ratio of each patch to encourage the model to focus on capturing fine details and essential features. We refer the reader to the Supplementary Material regarding the details of the patches selection implementation.

## 3.2 Training Data Generation

In this section, we introduce our approach for generating large-scale synthetic data and curating high-quality real-world data.

### 3.2.1 Synthetic Data Generation

Our synthetic data generation is based on rigged human assets, where we articulate human shapes by applying real-world human poses sampled from the AMASS dataset Mahmood et al. (2019). In this subsection, we first introduce the preliminaries of the SMPL-X model. Then we formulate our strategy for data generation.

**Parametric body model.** Parametric models like SMPL Loper et al. (2015) and SMPL-X Pavlakos et al. (2019) provide a mapping from pose and shape parameters to the mesh surface. For example, SMPL-X Pavlakos et al. (2019) is defined as a mapping function $M(\boldsymbol{\theta}, \boldsymbol{\beta}, \boldsymbol{\psi}) : \mathbb{R}^{|\boldsymbol{\theta}|} \times \mathbb{R}^{|\boldsymbol{\beta}|} \times \mathbb{R}^{|\boldsymbol{\psi}|} \rightarrow \mathbb{R}^{3N}$, where $\boldsymbol{\theta}, \boldsymbol{\beta}, \boldsymbol{\psi}$ are the parameters for pose, shape, and facial expression, respectively. The function of SMPL-X is formulated as follows:

$$\mathbf{M}(\boldsymbol{\beta}, \boldsymbol{\theta}, \boldsymbol{\psi}) = W(\mathbf{T}(\boldsymbol{\beta}, \boldsymbol{\theta}, \boldsymbol{\psi}), J(\boldsymbol{\beta}), \boldsymbol{\theta}, \mathbf{W}), \tag{5}$$

$$\mathbf{T}(\boldsymbol{\beta}, \boldsymbol{\theta}, \boldsymbol{\psi}) = \bar{\mathbf{T}}_m + B_S(\boldsymbol{\beta}) + B_E(\boldsymbol{\psi}) + B_P(\boldsymbol{\theta}), \tag{6}$$

where $B_P(\cdot)$, $B_S(\cdot)$ and $B_E(\cdot)$ denote pose, shape, and expression blend functions, respectively, while $\mathbf{W}$ is the associated set of blend weights. The corrective blend shapes, *i.e.,* $B_P(\boldsymbol{\theta})$, $B_E(\boldsymbol{\psi})$ and $B_S(\boldsymbol{\beta})$, supply per-vertex offsets that deform the template human mesh $\bar{\mathbf{T}}_m$. Subsequently, linear blend skinning $W(\cdot)$ rotates these deformed vertices around the joints $J(\boldsymbol{\beta})$ and smooths the transformations via blend weights $\mathbf{W}$, returning the final human mesh.

**Synthetic data generation procedure.** During synthetic data generation, we first randomly sample a set of SMPL-X parameters from the AMASS Mahmood et al. (2019) dataset $\{\mathbf{R}^{\text{src}}, \mathbf{T}_{\text{src}}\} \sim \mathcal{D}_{\text{AMASS}}$. After that, we utilize the above function to animate the rigged human asset $\mathbf{M}$, re-center $RC(\cdot)$ the animation results, and assign a set of camera viewpoints $\mathbf{C}$ for rendering multiple views to serve as training data,

$$\{\mathbf{I}_i\}_{i=1}^N = \text{Render}(\text{A}(RC(\mathbf{M}), \{\mathbf{R}^{\text{src}}, \mathbf{T}_{\text{src}}\}), \{\mathbf{C}_i\}_{i=1}^N). \tag{7}$$

### 3.2.2 Real-world Data Generation

We build our fitting pipeline for data generation to enable arbitrary novel views rendering from multi-camera captured data, which could serve as supervision to train our framework. This pipeline is built upon the 3D Gaussian Splatting (3DGS) representation Kerbl et al. (2023), which is particularly suitable for fitting because of its fast optimization and real-time rendering capability. Specifically, 3DGS represents a scene as a set of 3D Gaussians, each defined by a center position $p$ and a covariance matrix $\Sigma$, capturing both spatial location and shape:

$$G(x) = \exp\left(-\frac{1}{2}(x-p)^T \Sigma^{-1}(x-p)\right), \tag{8}$$

where $x$ is a point in 3D space. These Gaussians are projected onto the image plane and composited using a tile-based rasterizer with alpha blending, supporting differentiable and real-time rendering.

Given a set of images $\mathcal{I} = \{I_i \mid i \in [1, M]\}$ captured from multiple-view datasets *e.g.,* DNA-Rendering Cheng et al. (2023) and MVHumanNet Xiong et al. (2024)) and the corresponding SMPL-X mesh of the human subject, we initialize the 3D Gaussians by assigning one Gaussian to each mesh vertex. The center of each Gaussian is set to the vertex position, and its covariance is initialized accordingly. We then optimize the parameters of these Gaussians by minimizing the photometric reconstruction loss:

$$\mathcal{L} = \left\| I_i - f\left(V(\theta), \pi_i\right) \right\|^2, \tag{9}$$

where $V(\theta)$, $\pi_i$, and $f(\cdot)$ denote the set of Gaussian parameters, the camera parameters and the differentiable rendering function, respectively. During training, we sample one view per iteration and apply adaptive density control Kerbl et al. (2023) to improve convergence and coverage.

After fitting, we obtain a refined 3D Gaussian representation of the posed human, which we denote with $G_p$. Finally, we render multiple images from the re-centered $G_p$ under a predefined set of canonical viewpoints, which could serve as a data source for training.

## 4 EXPERIMENTS

In this section, we present our experimental evaluation. We begin by describing our experimental setup, which includes the datasets, implementation details, and evaluation metrics. Then, we compare our method with state-of-the-art approaches both quantitatively and qualitatively. Finally, we conduct ablation studies for the most important components of our approach.

### 4.1 EXPERIMENTAL SETUP

**Implementation details.** Our framework is implemented on PyTorch. The experiments are conducted on NVIDIA H100, and we utilize 64 H100s for training. The hyperparameters $\lambda_m$ and $\lambda_p$ are both set to 0.5. During pre-training, we set $N = 4$ as the number of rendered views. We utilize AdamW as the optimizer. The learning rate is set as 6e-4 with the batch size as 64. The triplane spatial size in this work is 96. The cropped patch size during training is 180.

**Datasets.** During training, we utilize two kinds of data sources, *i.e.,* commonly utilized human scans (THuman2 Yu et al. (2021), CustomHuman Shen et al. (2023), and 2K2K Han et al. (2023)) and our generated data. The former size contains around 5k assets with 180k images, while the latter one has 100k assets with 2.6M images. Evaluation is performed on three benchmarks, including THuman2, CustomHuman, and 2K2K datasets.

**Evaluation metrics.** Following previous works Pan et al. (2024); AlBahar et al. (2023); He et al. (2025), we evaluate the reconstructed avatar on both rendering and geometry quality. During evaluation of rendering quality, we select three widely-used metrics, *i.e.,* Peak Signal-to-Noise Ratio (PSNR), Structural Similarity Index Measure (SSIM) Wang et al. (2004), and Learned Perceptual Similarity (LPIPS) Zhang et al. (2018). For a fair comparison, the resolution of rendered images is set to $512 \times 512$. Our rendered views are uniformly distributed around the human subject using 20-degree intervals. For geometry evaluation, we report Chamfer Distance (CD), Normal Consistency (NC), and F-Score.

### 4.2 COMPARISON WITH STATE-OF-THE-ART METHODS

**Baselines.** The baselines we use in our evaluation include both methods for general object reconstruction and methods for human-specific reconstruction from a single image. Specifically, Real3D Jiang et al. (2025) and SF3D Boss et al. (2025) are LRM-based state-of-the-art methods in general object reconstruction, which can achieve fast reconstruction with a single feed-forward inference. PaMIR Zheng et al. (2021), SiFU Zhang et al. (2024b) and SiTH Ho et al. (2024) require instance-level post-processing to return the 3D human, usually involving techniques like diffusion, mesh fitting, *etc.* However, these stages lead to a significant increase in the inference time. Trellis Xiang et al. (2025) and Hunyuan2 Zhao et al. (2025) are two very recent large-scale industrial foundation models that perform flow-based generative modeling in 3D space, trained on around 1 million diverse 3D assets to achieve high-quality generation.

**Evaluation Protocol.** It follows the established practice in this field: each method is trained using the settings from its original paper, and all methods are then evaluated on the same benchmark dataset for comparison. This is consistent with prior works (e.g., SiTH, SiFU, Trellis, Hunyuan2) and allows a fair comparison of generalization ability and reconstruction quality.

The experiments are conducted on multiple benchmarks *i.e.,* CustomHuman Shen et al. (2023), THuman2 Yu et al. (2021) and 2K2K Han et al. (2023). Existing benchmarks are built based on normalized human scans, such that the canonical view is defined as the subject's frontal perspective. However, in real-world scenarios, it is not guaranteed that the input images will always show a frontal view. To address this discrepancy, we additionally incorporate side-view data into our experimental evaluations, which helps assess the robustness and generalizability of our approach. Specifically, we render images from a viewpoint that has a horizontal offset of 60 degrees relative to the canonical view and use them as input. Considering that some methods (SiTH, SiFU, Trellis, and Hunyuan2) output mesh as the final representations, we adopt a fair comparison protocol by first aligning the scale of the reconstructed meshes to that of the ground truth. Subsequently, we apply an Iterative Closest Point (ICP) alignment to further refine the correspondence between the reconstructions and the ground truth. Finally, we utilize the re-aligned mesh for rendering and geometry evaluation.

Table 1: **Comparison with previous state-of-the-art methods on rendering quality.** These include Real3D, SF3D, Trellis, Hunyuan2, SiFU and SiTH, on the CustomHuman, THuman2 and 2K2K datasets. We outperform all previous methods across all evaluated metrics with a notable gain. ↑ and ↓ represent the higher the better, and the lower the better, respectively.

| Method | CustomHuman | | | THuman2 | | | 2K2K | | |
|---|---|---|---|---|---|---|---|---|---|
| | PSNR↑ | SSIM↑ | LPIPS↓ | PSNR↑ | SSIM↑ | LPIPS↓ | PSNR↑ | SSIM↑ | LPIPS↓ |
| *Input: Frontal view* | | | | | | | | | |
| Real3D Jiang et al. (2025) | 17.13 | 0.8990 | 95.12 | 19.14 | 0.9094 | 87.68 | 18.06 | 0.9020 | 81.78 |
| SF3D Boss et al. (2025) | 19.46 | 0.9113 | 66.09 | 22.28 | 0.9287 | 57.20 | 20.47 | 0.9142 | 58.14 |
| Trellis Xiang et al. (2025) | 18.59 | 0.9123 | 74.98 | 20.77 | 0.9218 | 65.67 | 19.21 | 0.9140 | 68.25 |
| Hunyuan2 Zhao et al. (2025) | 19.42 | 0.9094 | 74.34 | 21.44 | 0.9257 | 66.19 | 19.87 | 0.9145 | 65.62 |
| PaMIR Zheng et al. (2021) | 18.15 | 0.9070 | 88.12 | 21.03 | 0.9229 | 70.91 | 18.89 | 0.9113 | 73.90 |
| SiFU Zhang et al. (2024b) | 17.94 | 0.9091 | 85.75 | 19.44 | 0.9157 | 79.62 | 16.82 | 0.9039 | 87.51 |
| SiTH Ho et al. (2024) | 19.13 | 0.9173 | 72.94 | 20.92 | 0.9231 | 66.90 | 18.49 | 0.9095 | 73.55 |
| SAGE | **22.29** | **0.9360** | **42.42** | **23.96** | **0.9382** | **42.13** | **22.65** | **0.9336** | **41.72** |
| *Input: Side view* | | | | | | | | | |
| Real3D Jiang et al. (2025) | 17.42 | 0.9005 | 94.94 | 19.40 | 0.9109 | 87.90 | 18.67 | 0.9047 | 78.54 |
| SF3D Boss et al. (2025) | 19.05 | 0.9085 | 69.75 | 21.25 | 0.9227 | 65.02 | 20.50 | 0.9139 | 57.85 |
| Trellis Xiang et al. (2025) | 17.21 | 0.9071 | 88.08 | 19.50 | 0.9172 | 74.79 | 18.12 | 0.9088 | 74.57 |
| Hunyuan2 Zhao et al. (2025) | 19.22 | 0.9076 | 77.28 | 21.01 | 0.9234 | 70.02 | 19.81 | 0.9139 | 66.48 |
| PaMIR Zheng et al. (2021) | 17.24 | 0.9006 | 97.10 | 20.59 | 0.9190 | 76.41 | 18.27 | 0.9077 | 78.95 |
| SiFU Zhang et al. (2024b) | 17.25 | 0.9040 | 93.26 | 19.32 | 0.9145 | 82.28 | 17.24 | 0.9052 | 85.49 |
| SiTH Ho et al. (2024) | 18.25 | 0.9114 | 81.95 | 20.43 | 0.9201 | 71.92 | 18.72 | 0.9102 | 73.25 |
| SAGE | **22.52** | **0.9348** | **43.58** | **24.35** | **0.9382** | **42.55** | **23.07** | **0.9347** | **40.95** |

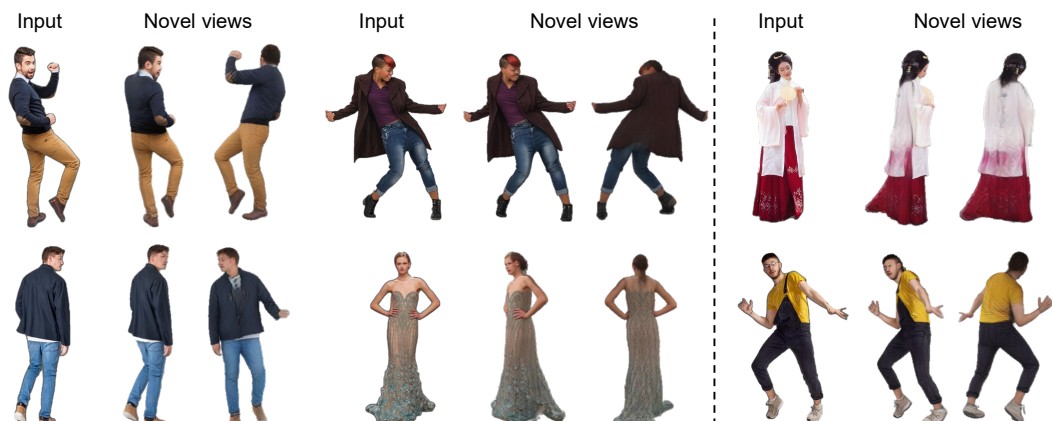

Figure 3: **Qualitative evaluation of our approach.** Results of SAGE on in-the-wild images. We also show some typical failure cases (right), *e.g.,* inferring the plausible back texture of challenging clothes like dresses and overalls. (Best viewed in color.)

**Results.** As shown in Table 1, we observe that our method outperforms all previous approaches under different input conditions. Compared with the best competitor SiTH, our method achieves 41.8%, 37.0% and 43.3% relative LPIPS gain on the CustomHuman, THuman2 and 2K2K datasets, respectively. It is evident that previous general object reconstruction methods like Real3D, SF3D and Hunyuan2 are moderately robust to variations in viewpoint. However, they suffer from suboptimal performance due to the lack of specialized training on human-specific data. Additionally, previous state-of-the-art human reconstruction approaches often yield lower quality results when given a side-view input, as they do not generalize well to such less common viewpoints. In contrast, our approach yields superior performance in both appearance and structure. For geometry evaluation (Table 2), SAGE consistently achieves the best performance across all metrics with a notable gain. Notably, on the side-view inputs, where occlusion affects performance, SAGE outperforms human-specific method SiTH by 94.3% relative F-Score gain on CustomHuman.

The qualitative comparison in Figure 4 also validates the effectiveness of our method. Beyond the results presented in Figure 1, we also showcase results on a diverse set of in-the-wild images in Figure 3. These images include a wide range of poses and clothing styles, demonstrating that our method effectively captures fine details and accurately represents the 3D structure even in challenging conditions. However, we also observe certain failure cases; for example, the network

Table 2: **Comparison with previous state-of-the-art methods on geometry quality.** The evaluation is conducted on the CustomHuman, THuman2 and 2K2K datasets. When reporting CD, we consider both the prediction to ground truth and the ground truth to prediction distance. We outperform all previous methods across all evaluated metrics with a notable gain. ↑ and ↓ represent that higher is better, and that lower is better, respectively.

| Method | CustomHuman | | | THuman2 | | | 2K2K | | |
|---|---|---|---|---|---|---|---|---|---|
| | CD↓ | NC↑ | F-Score↑ | CD↓ | NC↑ | F-Score↑ | CD↓ | NC↑ | F-Score↑ |
| *Input: Frontal view* | | | | | | | | | |
| SF3D Boss et al. (2025) | 1.738/2.040 | 0.847 | 39.585 | 1.441/1.745 | 0.833 | 43.820 | 1.204/1.412 | 0.829 | 50.900 |
| Trellis Xiang et al. (2025) | 2.125/2.175 | 0.801 | 32.846 | 1.799/1.832 | 0.796 | 37.939 | 1.446/1.359 | 0.805 | 48.826 |
| Hunyuan2 Zhao et al. (2025) | 1.799/1.762 | 0.837 | 38.365 | 1.562/1.541 | 0.808 | 43.868 | 1.237/1.217 | 0.829 | 53.946 |
| ICON Xiu et al. (2022) | 2.468/2.915 | 0.779 | 27.731 | 2.568/3.168 | 0.752 | 26.453 | 2.211/3.331 | 0.728 | 28.805 |
| ECON Xiu et al. (2023) | 2.160/2.813 | 0.804 | 33.429 | 2.240/3.931 | 0.763 | 31.294 | 2.066/6.232 | 0.732 | 32.927 |
| SiFU Zhang et al. (2024b) | 2.440/3.203 | 0.789 | 27.553 | 2.509/3.778 | 0.760 | 27.487 | 2.136/5.331 | 0.732 | 29.823 |
| SiTH Ho et al. (2024) | 1.792/2.215 | 0.826 | 36.822 | 1.741/2.082 | 0.805 | 39.666 | 1.518/1.896 | 0.798 | 42.859 |
| SAGE | **1.062/1.102** | **0.867** | **61.379** | **1.027/1.098** | **0.840** | **61.939** | **1.045/1.110** | **0.836** | **60.673** |
| *Input: Side view* | | | | | | | | | |
| SF3D Boss et al. (2025) | 1.657/2.177 | 0.834 | 39.931 | 1.695/2.284 | 0.810 | 39.102 | 1.234/1.452 | 0.823 | 49.584 |
| Trellis Xiang et al. (2025) | 2.361/4.802 | 0.757 | 29.603 | 2.089/2.933 | 0.761 | 34.754 | 1.606/1.644 | 0.772 | 44.555 |
| Hunyuan2 Zhao et al. (2025) | 1.848/1.898 | 0.827 | 37.698 | 1.642/1.677 | 0.811 | 42.618 | 1.259/1.235 | 0.821 | 54.622 |
| ICON Xiu et al. (2022) | 2.536/3.635 | 0.769 | 28.138 | 2.286/3.012 | 0.760 | 30.981 | 1.892/2.873 | 0.745 | 35.828 |
| ECON Xiu et al. (2023) | 2.592/5.233 | 0.756 | 27.284 | 2.270/4.747 | 0.750 | 31.298 | 1.811/5.778 | 0.739 | 36.933 |
| SiFU Zhang et al. (2024b) | 2.615/6.112 | 0.756 | 24.788 | 2.301/4.631 | 0.748 | 30.057 | 1.820/5.550 | 0.736 | 35.653 |
| SiTH Ho et al. (2024) | 2.063/3.219 | 0.793 | 32.189 | 1.849/2.557 | 0.789 | 36.070 | 1.633/1.879 | 0.791 | 40.807 |
| SAGE | **1.052/1.079** | **0.858** | **62.553** | **1.044/1.123** | **0.829** | **62.344** | **1.058/1.133** | **0.829** | **60.126** |

Table 3: Ablation on the generated data type on the CustomHuman dataset.

Table 4: Ablation on the generated data scale on the CustomHuman dataset.

Table 5: Ablation on the impact of model settings on the CustomHuman dataset.

| Method | PSNR↑ | SSIM↑ | LPIPS↓ |
|---|---|---|---|
| w/o gen-data (assets) | 21.84 | 0.9333 | 46.51 |
| w/o gen-data (multi-cam) | 21.76 | 0.9326 | 47.83 |
| SAGE | **22.07** | **0.9344** | **45.18** |

| Method | PSNR↑ | SSIM↑ | LPIPS↓ |
|---|---|---|---|
| 25% | 21.98 | 0.9313 | 50.14 |
| 50% | 22.02 | 0.9338 | 47.03 |
| SAGE | **22.07** | **0.9344** | **45.18** |

| Method | PSNR↑ | SSIM↑ | LPIPS↓ |
|---|---|---|---|
| w/o mesh prior | 21.89 | 0.9334 | 46.26 |
| small triplane size (32) | 21.78 | 0.9323 | 48.33 |
| SAGE | **22.07** | **0.9344** | **45.18** |

sometimes struggles to accurately infer plausible textures on the back of the subjects. Addressing these shortcomings may require the adoption of more powerful models in future work.

## 4.3 ABLATION STUDIES

We perform ablation studies to highlight the most important components in our proposed method, including data generation and model settings. Unless stated otherwise, training runs for 70% of the total iterations adopted in the main comparison. The evaluation is conducted on the CustomHuman dataset with the frontal view as the input.

**Effectiveness of our generated data.** Our generated data includes two complementary components: *gen-data (assets)*, i.e., data generated from rigged assets, and *gen-data (multi-cam)*, which is generated from fitting real-world multi-camera data. As shown in Table 3, both components contribute effectively to the training process, enhancing the robustness and diversity of the model. Notably, synthetic assets can provide extensive diversity on the pose, given the animation strategy. Besides, the real-world data, though smaller in quantity, provides important signals for achieving high-quality reconstruction, especially in terms of photorealism. We believe this is due to the natural appearance statistics and complex visual cues present in real-world images, which are difficult to fully simulate through synthetic data alone. To further analyze the impact of data scale, we vary the portion of generated data used during training. As shown in Table 4, model performance improves consistently with more generated data.

**Impact of model settings.** We begin by investigating the impact of incorporating a mesh prior into our reconstruction pipeline (Table 5). When integrated, the mesh prior yields a 2.3% relative improvement in LPIPS, demonstrating that, even with a large-scale training dataset, mesh provides a valuable cue for capturing detailed human structural features. In addition, we adopt the triplane representation as the final 3D depiction, which compresses 3D geometry by decomposing the volume into three orthogonal projection planes. As shown in Table 5, reducing the triplane spatial size to 32, a common setting in LRMs, results in a significant performance drop. This finding highlights the

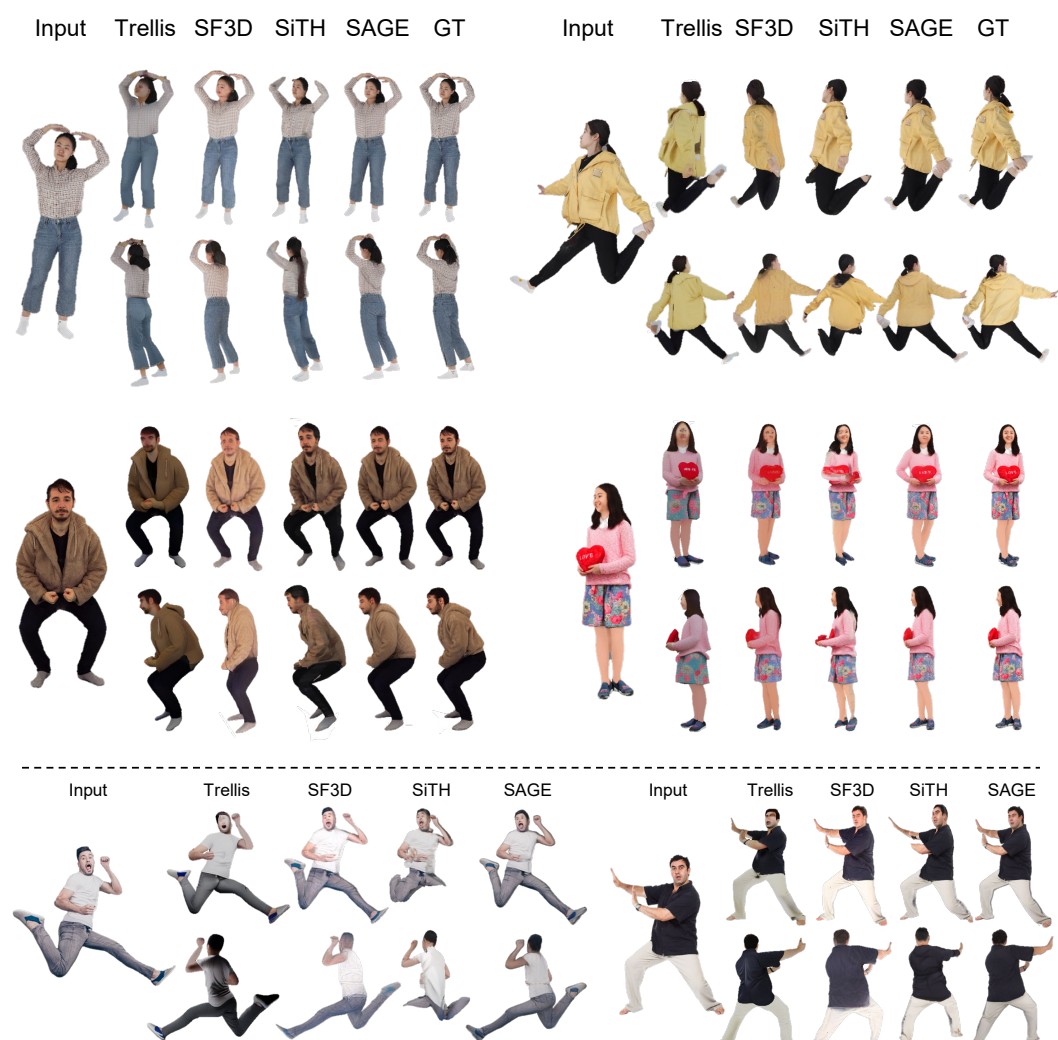

Figure 4: **Qualitative comparison with state-of-the-art methods**, including input from benchmarks (top) and in-the-wild images (bottom). The reconstructed human by our SAGE method shows superior structure and appearance. (Best viewed in color.)

necessity for a model capacity that aligns with the demanding requirements of human reconstruction, where capturing fine-grained details is more critical compared to general object reconstruction.

## 5  CONCLUSION

In this work, we present SAGE, a large human reconstruction model that enables generalizable and photorealistic 3D single-view human reconstruction in less than a second. Our contributions are two-fold, focusing on data generation and model design. To mitigate the scarcity of 3D human data, we design an effective and scalable data generation pipeline that integrates multiple sources (e.g., 3D scans and multi-camera captures), expanding the training dataset to 100k assets and significantly improving its diversity and scale. Built on this large-scale data, SAGE leverages the estimated simplified human mesh as an extra prior and directly lifts the input single image to a triplane representation that effectively captures the details of the 3D human. Extensive experiments on three benchmark datasets show that our approach consistently outperforms existing methods, with over 40% improvement in LPIPS and more than 90% relative gain in F-Score. We hope our designed framework could inspire future research into complex real-world scenarios, including dynamic clothing, occlusion, and rich human interactions.

**Ethics statement.** This work focuses on fast and photorealistic 3D human reconstruction from a single image. While such advancements hold significant promise, they also raise concerns about potential misuse, including deceptive practices, harassment, and privacy violations. The lowered barrier to creating realistic human reconstructions may facilitate harmful or unauthorized content, intensifying privacy risks. These considerations underscore the importance of responsible development, as well as appropriate technical and regulatory safeguards.

**Reproducibility statement.** We have provided the details of the experiment setup in Section 4.1 and Section C, including implementation details, benchmarks and evaluation metrics. We will release our generated data, generation pipeline, and our model upon acceptance of this paper.

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

## A  APPENDIX

In this Supplementary Material, we provide additional details that were not included in the main submission due to space constraints. Specifically, we include more discussions (Section B), more details on the experiment setup (Section C), and more visual results (Section D).

## B  MORE DISCUSSIONS

**Quality of our generated data.** Our generated dataset contains two components, *i.e.,* the synthetic subset and the real-world subset. For the synthetic subset, we render synthetic multi-view images in Blender from rigged SynBody Yang et al. (2023) assets. Because SynBody meshes and textures are of high quality and Blender's renderer preserves their visual fidelity, the resulting images maintain the same level of detail and realism as the source assets. For the real-world subset, this subset is generated by fitting multi-view capture data to 3DGS. We then re-render the fitted 3DGS back to the capture viewpoints and compare them with the corresponding ground-truth images, achieving an average of 36.23 / 0.9881 / 16.57 (PSNR / SSIM / LPIPS).

**Design of real-world data generation.** We argue that good initialization is critical in the real-world data generation stage. COLMAP initialization is the common practice in 3DGS, and it performs poorly on multi-view human capture data. To illustrate, we tested them on 10 randomly selected samples. Our improved initialization based on SMPL-X vertices enables significantly shorter training iterations, reducing optimization time from 40 minutes to 4 minutes. The performance on the initialization method is shown in Table 6.

Table 6: **Impact of initialization in real-world data generation.** ↑ and ↓ represent the higher the better, and the lower the better, respectively.

| Method | PSNR↑ | SSIM↑ | LPIPS↓ |
|---|---|---|---|
| COLMAP | 16.49 | 0.9025 | 68.79 |
| Ours | **36.38** | **0.9886** | **16.36** |

**Effectiveness of our generated data.** We further provide evidence that could validate the effectiveness of our generated data via fine-tuning Real3D Jiang et al. (2025) (a general 3D object reconstruction method) on our proposed dataset. As shown in Table 7, it can be observed that Real3D could benefit from our proposed new data with a notable performance gain on multiple benchmarks.

Table 7: **Effectiveness of our generated data**. ↑ and ↓ represent the higher the better, and the lower the better, respectively.

| Method | CustomHuman | | | THuman2 | | | 2K2K | | |
|---|---|---|---|---|---|---|---|---|---|
| | PSNR↑ | SSIM↑ | LPIPS↓ | PSNR↑ | SSIM↑ | LPIPS↓ | PSNR↑ | SSIM↑ | LPIPS↓ |
| *Input: Frontal view* | | | | | | | | | |
| Real3D | 17.13 | 0.8990 | 95.12 | 19.14 | 0.9094 | 87.68 | 18.06 | 0.9020 | 81.78 |
| Real3D (+ our data) | **20.97** | **0.9268** | **58.54** | **23.10** | **0.9325** | **55.30** | **20.91** | **0.9202** | **58.22** |
| *Input: Side view* | | | | | | | | | |
| Real3D | 17.13 | 0.8990 | 95.12 | 19.14 | 0.9094 | 87.68 | 18.06 | 0.9020 | 81.78 |
| Real3D (+ our data) | **19.46** | **0.9113** | **66.09** | **22.28** | **0.9287** | **57.20** | **20.47** | **0.9142** | **58.14** |

**Comparison with animation-based methods.** Methods like SHERF Hu et al. (2023) and LHM Qiu et al. (2025) fall under a related but distinct task setting: they aim to build generalizable, animatable human avatars from a single image. Since these methods focus primarily on pose-driven animation, they typically rely on ground-truth, high-quality SMPL/SMPL-X poses during both training and testing to ensure precise alignment.

In contrast, our task operates under a more challenging input condition: only a single image is available. That is to say, SMPL(-X) parameters must be estimated using off-the-shelf tools. For SHERF and LHM, the SMPL(-X) pose is derived from 4D-Humans (same as our method) and its provided demo, respectively. We follow SHERF's instructions from Github repo and train it on

the corresponding dataset. For LHM, since it does not release training and testing code, so we directly utilize its pre-trained weight and its demo code to perform inference. As shown in Table 8, it can be observed that our method achieves superior performance with notable gains across multiple datasets and input viewpoints. For methods (LHM/SHERF), they need to animate the avatar from the canonical space to the target pose space, during which any pose misalignment will directly exist in the rendering results, leading to performance degradation.

Table 8: **Comparison with animation-based methods.** ↑ and ↓ represent the higher the better, and the lower the better, respectively.

| Method | PSNR↑ | SSIM↑ | LPIPS↓ |
|---|---|---|---|
| SHERF Hu et al. (2023) | 16.83 | 0.9037 | 87.99 |
| LHM Qiu et al. (2025) | 17.75 | 0.9083 | 76.85 |
| Ours | **22.29** | **0.9360** | **42.42** |

**Future work.** In this work, our proposed framework advances the state-of-the-art in human reconstruction, achieving superior quality and faster inference speeds compared to previous methods. It is notable that some other factors are worth investigating to improve the capability of our framework, *e.g.,* extending reconstruction to human-interaction or generating the avatar under the text prompt.

## C  MORE EXPERIMENT SETUP DETAILS

**Our data generation.** For synthetic data generation, we utilize all 1,000 unique characters officially released by SynBody Yang et al. (2023) for public download. These assets exhibit high diversity in terms of ethnicity, body shape, and clothing. Specifically, the characters span a wide range of skin tones, and wear diverse outfits based on approximately 68 clothing templates, including dresses, T-shirts, coats, pants, and more. For real-world data generation, the Gaussian optimization only lasts 4,000 iterations with the densification performed between iterations 400 and 1,500. For other parameters, we follow the original settings Kerbl et al. (2023). Note that both data generation strategies are scalable. The data size of the generated real-world data is 22k assets, while the generated synthetic data contains 78k assets. We generate an average of 26 views per asset, with camera positions randomly distributed on a sphere. The azimuth angles range from $0°$ to $360°$, and the elevation angles range from $-45°$ to $60°$. These views are randomly sampled without enforcing any preference for frontal views during the training of SAGE.

**SAGE traning.** The input image resolution of SAGE is 512×512. The NeRF Mildenhall et al. (2020) MLP contains 10 layers with the width set as 60 and SiLU Elfwing et al. (2018) is utilized as the activation function. The number of samples per ray is set as 128. For datasets with 3D scans such as THuman, CustomHuman, and 2K2K, we follow a unified preprocessing pipeline. We place each 3D mesh under a canonical camera setup and render 36 multi-view images at 10-degree intervals along a horizontal 360° circle. These rendered RGB images are then used as the supervision signals during training. Importantly, we do not use the original 3D meshes themselves for supervision, but only the rendered images.

**Patch selection.** We utilize patches from the image to provide the supervision signals. Its selection process is formulated as follows. Given an image $I \in \mathbb{R}^{H \times W \times C}$ and a corresponding binary mask $M \in \{0, 1\}^{H \times W}$, we aim to sample a square patch of size $r \times r$ (e.g., $r = 180$) such that the region contains a sufficient proportion of foreground pixels. Let $R_{(i,j)} \subseteq M$ denote a square region of size $r \times r$ with its top-left corner at position $(i, j)$. The foreground ratio of region $R_{(i,j)}$ is computed as:

$$\alpha_{(i,j)} = \frac{1}{r^2} \sum_{u=0}^{r-1} \sum_{v=0}^{r-1} M_{i+u,j+v}.$$

We define the candidate set of regions as: $\mathcal{S} = \{(i,j) \mid \alpha_{(i,j)} \geq \tau, \ i,j \in [0, H-r] \text{ stride } s\}$. Here, $\tau$ is the mask threshold (e.g., $\tau = 0.05$), and $s$ is the stride used to slide the sampling window (e.g., $s = 10$). For each valid region $(i,j) \in \mathcal{S}$, we assign a sampling weight proportional to the number of foreground pixels,

$$w_{(i,j)} = \sum_{u=0}^{r-1} \sum_{v=0}^{r-1} M_{i+u,j+v}.$$

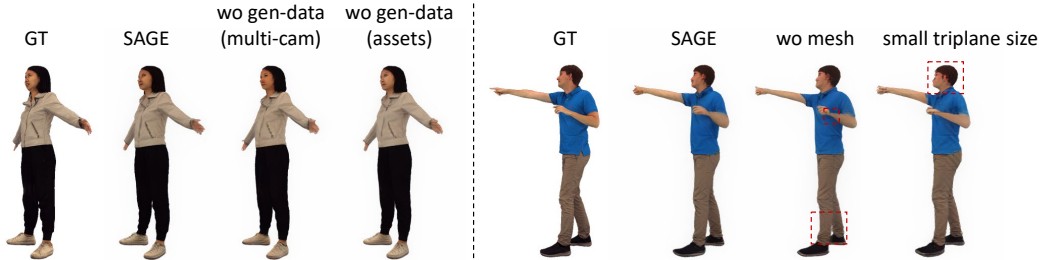

Figure 5: **Visualization of ablation studies on generated data type and model design.** Ablation on the training data and model settings are shown in the left and right parts, respectively. (Better viewed in color.)

The weights are normalized into a valid probability distribution:

$$p_{(i,j)} = \frac{w_{(i,j)}}{\sum_{(i',j')\in\mathcal{S}} w_{(i',j')}}.$$

Finally, a region $(i^*, j^*) \sim p$ is sampled, and the corresponding image patch is extracted as $P = I[i^* : i^* + r, \ j^* : j^* + r, \ :]$. This strategy encourages sampling of patches rich in foreground content while preserving diversity.

**Evaluation.** 50 assets from each dataset (CustomHuman, Thuman2, and 2K2K) are utilized for evaluation. Both CustomHuman and THuman2 feature diverse clothing styles, including loose garments and layered outfits, as well as a wide range of body poses, making them particularly challenging for accurate 3D reconstruction. In contrast, the 2K2K dataset primarily consists of humans in upright, standing poses. When performing 3D geometry evaluation, we extract the isosurface based on Marching Cubes Lorensen & Cline (1998) to convert SAGE's implicit representations into meshes. During evaluation, CD is measured in centimeters (cm), providing a precise indication of surface accuracy. F-Score is computed with a threshold of 0.01 meters.

# D MORE VISUAL RESULTS

We provide visualization of our ablation studies in Figure 5. The left part illustrates the impact of different training data configurations, while the right part shows the effects of varying model settings. Removing gen-data (multi-cam) leads to less photorealistic results, especially in fine appearance details. In contrast, removing gen-data (assets) weakens the model's capability to perceive and reconstruct human poses (see the right shoulder). For the right part, excluding the mesh prior degrades the structural quality of the output, while reducing the triplane resolution compromises the model's capacity to represent fine-grained details (see the red box).

As shown in Figure 6, we visualize our training data. The first three rows correspond to real-world generated data, while the remaining rows are generated synthetic data. Together, they provide the foundational training data that empowers SAGE to learn robust and generalizable 3D human representations.

To better showcase the results, we include additional in-the-wild reconstruction examples and comparisons with previous methods in the accompanying video file ("SAGE_Supp_Video.mp4"). In the video, the reconstruction results are presented with 360-degree rotation. From the video, it could be observed that our method is generally not affected by the Janus problem. This is because we model the 3D human directly as a whole in 3D space, instead of decomposing the task into separate front/back generation followed by heuristic merging, which is a common cause of inconsistent geometry or appearance across views (e.g., SiTH).

# E CLARIFICATION OF LLM USAGE

In this work, we employ LLMs to polish some sentences in the paper. Specifically, we provided draft sentences to the LLM, and asked it for advice regarding the word choice and sentence structure.

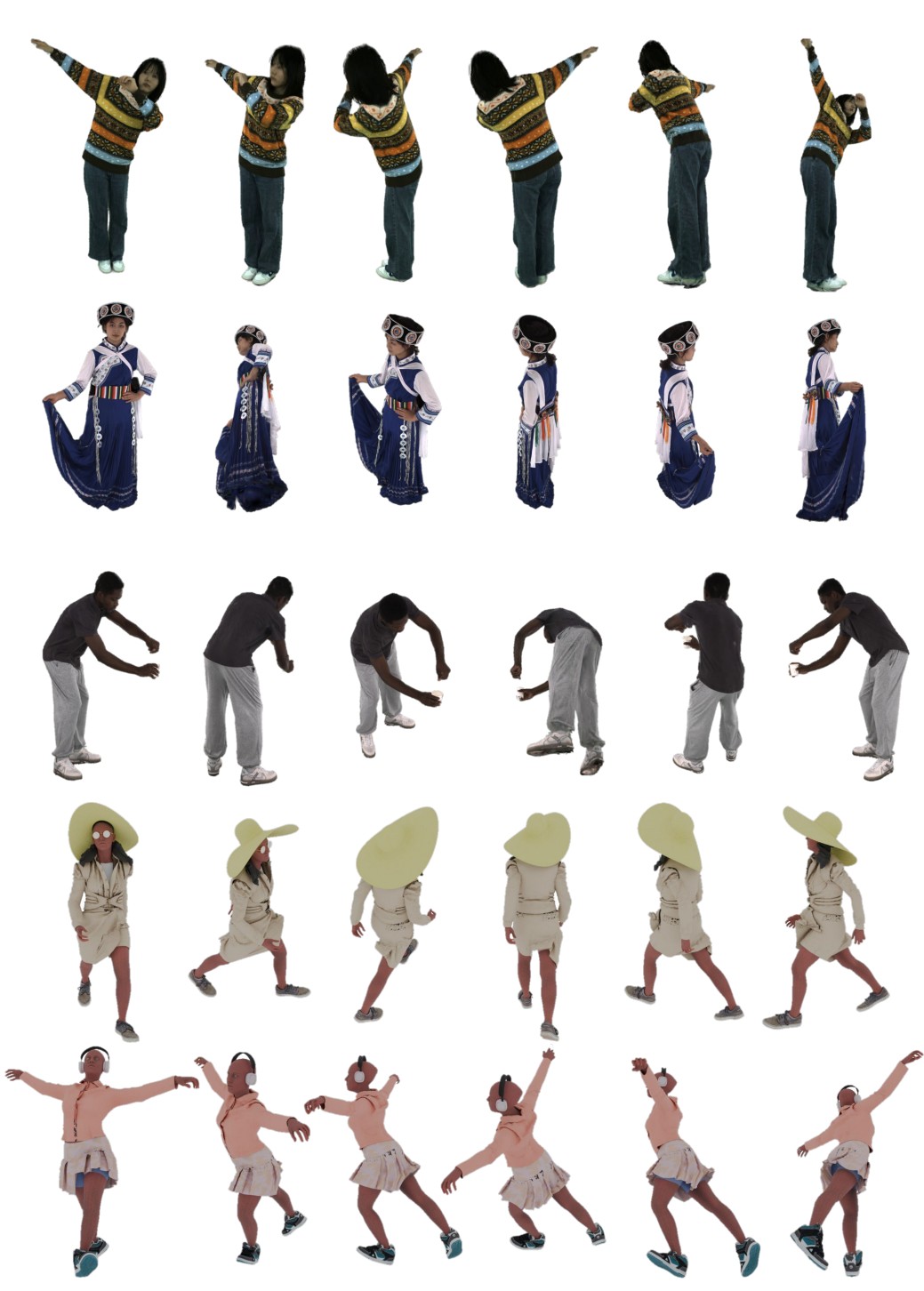

Figure 6: **Visualization of our training data.** (Best viewed in color.)

