# OpenReview forum: "SAGE: Fast, Generalizable and Photorealistic 3D Human Reconstruction from a Single Image"
_ICLR.cc/2026/Conference — ICLR 2026 Conference Withdrawn Submission_

### Official Review · Reviewer_3zUM · 2025-10-15

**Soundness:** 3
**Presentation:** 3
**Contribution:** 2
**Rating:** 2
**Confidence:** 5

**Summary:**

The paper proposes SAGE, a large-scale model for fast, photorealistic 3D human reconstruction from a single image. Its core contributions are a large-scale data generation pipeline (combining synthetic and real-world fitted data) and an LRM-based architecture variant that incorporates SMPL mesh priors. Experiments show significant improvements over existing methods across multiple metrics.

**Strengths:**

1. The construction of a 100K-asset training dataset is a substantial contribution with potential value to the community.
2. Sub-second reconstruction time (<1s) is a major practical advantage.
3. The paper includes extensive quantitative and qualitative comparisons across multiple benchmarks, along with ablation studies.

**Weaknesses:**

1. Table 1 may be misleading. The authors should provide a controlled experiment comparing SAGE against SiTH/SiFU (e.g., only the standard 5K public human scans). Otherwise, the reported ~40% LPIPS improvement is likely due to data scale rather than architectural superiority. The current results fail to demonstrate that SAGE’s design is more innovative than existing feed-forward human reconstruction models (e.g., LHM, HumanSplat).

2. The paper claims to introduce a *novel large human reconstruction model*, but in reality, it merely adapts the generic LRM framework to the human domain and relies heavily on massive data and computational resources. Core components (triplane representation, patch-level supervision, and SMPL priors) have already been widely used in prior works such as SHERF, Real3D, and SF3D. The authors do not clearly articulate the fundamental differences between SAGE and SHERF, nor do they prove that their architecture is more generalizable or scalable. If the 100K-data advantage is removed, SAGE’s performance gain likely vanishes. I believe treating data scaling as the only novelty is insufficient for acceptance at a top-tier conference.

3. I would like to see whether SAGE’s architectural design truly offers essential improvements over SHERF, which requires: 1) **Aligning training datasets** (e.g., train both models on the same 5K human scans), and 2) **Using the same image encoder** (e.g., both using DINOv2 or both using the same backbone).
Currently, it is highly plausible that SAGE’s gains stem not from its architecture, but from using a stronger image encoder (DINOv2) and/or vastly more training data.

**Questions:**

1. How does SAGE perform on completely unseen, out-of-domain test sets, such as HuMMan, ZJU-MoCap, or RenderPeople? Cross-domain generalization is the true test of robustness and practical utility.
2. After reconstruction, can SAGE support real-time rendering like LHM?

---

### Official Review · Reviewer_uyrR · 2025-10-26

**Soundness:** 3
**Presentation:** 3
**Contribution:** 2
**Rating:** 6
**Confidence:** 4

**Summary:**

As human datasets are limited, the paper proposes to augment the existing datasets for more data. The augmentation pipeline consists of two parts: 1) Synthetic dataset: It articulates the rigged human assets by poses provided by AMASS and renders from 3D. 2) Augmented real-world dataset: It fits 3DGS on multiview images and renders novel views.

For human reconstruction and rendering, it adopts the LRM-like architecture - a feed-forward network that outputs triplane of the 3D human subjects.

It outperforms existing baselines in terms of both reconstruction and rendering. Meanwhile, it works well on loose clothes, e.g., dresses and capes, and in-the-wild images.

**Strengths:**

* The paper is well-written and easy to follow. It describes the model architecture and data pipeline in sufficient details.
* It applies the model to in-the-wild images and loose clothes. This indicates the method's great ability of generalization.
* While collecting datasets from human subjects can be costly, implementing such a pipeline is valuable as it allows for model improvement at no/little additional cost.

**Weaknesses:**

The main contribution of this paper is a pipeline that processes and combines original datasets, thereby significantly scaling up the overall dataset size. For LRM-based models, both dataset scale and diversity are crucial for generalization. While the proposed pipeline effectively increases dataset scale, it does little to enhance diversity. I am interested in understanding how the method performs when trained on a simple combination of the original datasets used in the pipeline, without additional processing. For instance, one could train the model on the original multiview images from DNA-Rendering and MVHumanNet, along with the 3D assets in their canonical poses, without further articulation. Without such a baseline, it is difficult to assess the true effectiveness of the proposed data processing pipeline.

**Questions:**

Please see the weakness.

---

### Official Review · Reviewer_hvnz · 2025-10-28

**Soundness:** 3
**Presentation:** 3
**Contribution:** 3
**Rating:** 4
**Confidence:** 4

**Summary:**

The paper presents a fast and generalizable framework for photorealistic 3D human reconstruction from a single image. The paper introduces a scalable data generation pipeline combining synthetic and real-world multi-camera data, resulting in a 100K large-scale human dataset. The model achieves state-of-the-art performance with real-time inference and significant gains over previous single-image reconstruction methods in both rendering and geometric quality.

**Strengths:**

- Paper is well-written and easy to understand.
- The paper constructs a large-scale 3D human dataset (100K assets) by combining synthetic and multi-camera captures, substantially improving model robustness and generalization.
- The paper achieves significantly better performance than existing single-image reconstruction models while maintaining fast inference.

**Weaknesses:**

- The main concern is the absence of an ablation study without their generated dataset, i.e., trained only on publicly available datasets used by other methods, making it unclear how much of the performance gain stems from the model design itself rather than the newly constructed dataset. Moreover, since the model architecture appears relatively simple and closely follows prior LRM-style frameworks, the work lacks technical contribution beyond scaling data and model capacity.
- In Eq. (7), the term A is not defined. Clarification is needed on its origin and formulation.
- The paper does not specify the total training time or computational cost, which is essential for reproducibility.
- No evaluation or comparison on CAPE or RP datasets, which are one of the standard evaluation set for human reconstruction benchmarks. This would strengthen generalization claims.
- Missing recent references / comparisons, particularly on single-image reconstruction works such as:
    - Missing reference - [1] Dong et al., *“MoGA: 3D Generative Avatar Prior for Monocular Gaussian Avatar Reconstruction,”* ICCV 2025.
    - Missing reference - [2] Shin et al., *“CanonicalFusion: Generating Drivable 3D Human Avatars from Multiple Images,”* ECCV 2024.
    - Missing comparison - [3] Li et al., *“Pshuman: Photorealistic Single-Image 3D Human Reconstruction Using Cross-Scale Multiview Diffusion and Explicit Remeshing,”* CVPR 2025.

I will reconsider the score when all the concerns above the handled well.

**Questions:**

How does the model perform under challenging conditions such as occlusions or extreme lighting? Including visual examples of such cases would help readers understand the framework’s limitations.

---

### Official Review · Reviewer_QJbj · 2025-10-29

**Soundness:** 2
**Presentation:** 2
**Contribution:** 2
**Rating:** 2
**Confidence:** 4

**Summary:**

This paper presents SAGE, a single-view human reconstruction method that adapts the Large Reconstruction Model (LRM) for human-specific reconstruction. It introduces a data generation pipeline to produce large-scale synthetic human assets (about 100K assets). The approach integrates SMPL-based human priors into the LRM via token cross-attention-based mapping and demonstrates competitive results on three benchmarks.

**Strengths:**

- Large-scale data contribution: The generation of 100K human reconstruction dataset for training is a valuable resource. If released, this dataset could substantially benefit the community and future human reconstruction research.
- Integration of human priors: The paper effectively incorporates human-related priors into the LRM, adapting a general reconstruction model for the human domain.

**Weaknesses:**

#### 1. Experimental design and evaluation inconsistencies
- The quantitative comparison in Table 2 does not support the contribution of the generated dataset.
  The ablation studies (Tables 3–5) adopt totally different metrics and settings from Table 2, making it difficult to evaluate the influence of the synthetic data compared to other methods.
- The claim of state-of-the-art performance (Lines 99–102) is not well-supported.
  The improvements are mainly on LPIPS and F-Score, which are not the dominant metrics for 3D human reconstruction.
  On more representative metrics such as PSNR and Chamfer Distance (CD), the method does not achieve state-of-the-art performance.
- There is no consistent evaluation of novel-view synthesis (e.g., PSNR, SSIM, LPIPS) against other methods on major benchmarks (CustomHuman, THuman2, 2K2K).
  Only partial CustomHuman results appear in the ablation study without cross-method comparison.
- Some reported results are numerically inconsistent with prior work.
  For instance, in the paper of SIFU, the Chamfer Distance on THuman2 is 0.5961, while this paper reports 2.509 and 2.301, suggesting discrepancies in evaluation protocols or scaling.

#### 2. Fairness and completeness of comparisons
- Many baselines in Table 2 (e.g., Real3D, SF3D, Trellis, Hunyuan2) are general-purpose reconstruction models not fine-tuned for human datasets.  Without explicit clarification, the comparison may be unfair or misleading.
- Several recent, human-specific methods are missing from the quantitative comparison despite being discussed in related work:
  - Chen et al., 2025 (ICLR) – Generalizable Human Gaussians from Single-View Image, which achieves PSNR = 23.84, SSIM = 0.944 on CustomHumans, outperforming SAGE (PSNR = 22.07, SSIM = 0.9344).
  - Li et al., 2025 (CVPR) – PSHuman: Photorealistic Single-View Human Reconstruction using Cross-Scale Diffusion, another state-of-the-art, human-specific model.
  Omitting these comparisons weakens the fairness and completeness of the evaluation.

#### 3. Method
- The adaptation of LRM for human reconstruction, while practical, shows limited conceptual novelty.
  The method mainly integrates SMPL-based tokens with visual tokens in cross-attention, any other alternatives to verify the effectiveness of this integration? The impact of mesh prior is marginal reported in Table 5, as only 0.18 increase of PSNR and 0.001 in SSIM.

**Questions:**

See weakness part.

**Details Of Ethics Concerns:**

For the human data generation, there should be ethics review for bias, privacy.

---

### Note · Authors · 2025-11-14

I have read and agree with the venue's withdrawal policy on behalf of myself and my co-authors.